# Designing Activating Schoolyards: Seen from the Girls’ Viewpoint

**DOI:** 10.3390/ijerph16193508

**Published:** 2019-09-20

**Authors:** Charlotte S. Pawlowski, Jenny Veitch, Henriette B. Andersen, Nicola D. Ridgers

**Affiliations:** 1Department of Sports Science and Clinical Biomechanics, University of Southern Denmark, Campusvej 55, 5230 Odense M, Denmark; hbandersen@health.sdu.dk; 2Institute for Physical Activity and Nutrition (IPAN), School of Exercise and Nutrition Sciences, Deakin University, 221 Burwood Highway, Burwood, Victoria 3125, Australia; jenny.veitch@deakin.edu.au (J.V.); nicky.ridgers@deakin.edu.au (N.D.R.)

**Keywords:** recess physical activity, girls, schoolyard renovation, re-design, photo-elicitation interview, pen profile

## Abstract

Girls are typically less active in the schoolyard during recess than boys. It is therefore necessary to understand influences on girls’ recess activity in schoolyards. The aim of this qualitative study was to investigate girls’ perceptions of physical environmental factors influencing recess physical activity in re-designed schoolyards and to compare the perceptions of girls from different age groups. In 2018, 50 girls from five Danish schools were interviewed using photo-elicitation. The girls were from Grade 4 (*n* = 28, age 10–11) and Grade 6 (*n* = 22, age 12–13). Data were analysed using pen profiles constructed from verbatim transcripts. Ten factors emerged: variety, accessibility, size, designated spaces, greenery, playground markings, active play facilities, sports facilities, play equipment, and speakers. Play facilities (trampolines, obstacle courses, dancing and gymnastic appliances) were favoured over traditional sport facilities. Designated spaces, greenery and speakers were important for feeling comfortable within the schoolyard. Although similar factors were raised by the two age groups, some factors were perceived as enablers by the youngest and as barriers by the oldest girls, highlighting the complexity of designing schoolyards that cater to all ages. A greater understanding of how different designs and facilities may be perceived by girls of different ages is important for the design of future schoolyards.

## 1. Introduction

Physical activity (PA) is crucial for children’s health [1,2], but a large number of schoolchildren do not engage in the recommended minimum level of 60 minutes of moderate-to-vigorous-intensity physical activity (MVPA) per day in Denmark and other Western countries [3]. Girls are less likely to meet this recommendation than boys, with the latest Danish national survey indicating that 33% of boys and 19% of girls aged 11–15 years met the recommendation [4]. This highlights the need to identify ways to increase girls’ daily PA levels.

The schoolyard is a valuable setting for promoting children’s PA [5,6] as children from all backgrounds spend considerable time in the schoolyard during most, if not all, school days [7,8,9]. Time spent in the schoolyard during recess has been shown to contribute up to 40% of children’s recommended daily PA [8]. However, gender differences have been consistently found in recess PA, with boys engaging in more MVPA than girls [10,11]. During recess, girls typically prefer dyadic play and social interactions and tend to be less interested in competitive sport-based activities compared to boys [12,13]. Previous studies have also reported that boys often dominate schoolyards during recess, particularly on sporting grounds and ovals that often constitute a large part of the schoolyard [10,12,14]. This may limit opportunities for girls to be active unless they have a high level of skill and are therefore accepted by the boys to join the sport-based games [12,15,16,17,18]. These findings clearly demonstrate the need for schoolyard designs that encourage both boys and girls to be active [10,19,20,21,22].

A number of studies have modified schoolyards in order to increase PA during recess. Strategies used have included introducing playground markings and physical structures, providing unfixed equipment, staff training, and policy changes [23]. In general, these studies have shown increases in PA during recess, yet gender differences still remain, with boys observed to use schoolyards more than girls [24,25]. Therefore, it is important to examine girls’ perspectives on re-designed schoolyards to better understand what influences their activity engagement in these new schoolyards [24,26]. To our knowledge, only one study has exclusively explored young girls’ (8–10 years) perspectives on a schoolyard intervention that provided unfixed equipment to promote PA [27]. No studies have examined older girls’ perspectives. As schoolyards typically cater to children of a wide range of ages, it is important to determine whether specific features are viewed in the same way by girls of different ages [26]. 

The aim of this study was to investigate girls’ perceptions of physical environmental factors influencing recess PA in re-designed schoolyards and to compare the perceptions of girls from different age groups.

## 2. Materials and Methods

This study was nested within a large quasi-experimental schoolyard intervention study, the Activating Schoolyards Study, which investigated the impact of re-designed schoolyards on children’s PA during recess. The study protocol for the overall study was described in more depth elsewhere [28]. The current sub-study was conducted in 2018, three years after the schoolyard re-design in which schools were invited to participate in a qualitative follow-up study that examined girls’ perspectives concerning their re-designed schoolyard.

In accordance with social constructionism [29], the intention of this study was to listen to the children and examine ways in which the schoolyard, as a social phenomenon, was perceived and institutionalised by the girls. Allowing girls to voice their perspectives is important in order to understand them and improve their PA [30]. A research perspective supportive of this approach is ‘the new social studies of childhood’ [31]. From the late 1990s onwards, this new approach sought to recognise children as active agents, competent social actors and individuals with rights [31]. However, in medical and health science, research methods are predominantly adult-led and quantitative [32]. This may be valuable in identifying PA prevalence and associations. However, these methods do not provide contextual understanding and cannot readily explain why some children are more physically active than others [33,34].

### 2.1. Context

In Denmark, school is mandatory for children aged 6–16 years. Schools are typically organised into three tiers at the one location: junior (Grades 0–3, 6–9 years old), middle (Grades 4–6, 10–12 years old) and senior (Grades 7–10, 13–16 years old). The schoolyards are primarily used during allocated morning recess and lunchtime (from hereon called recess), which, combined, is approximately 60 min in total per day. Each school has their own policies relating to recess. For example, it is up to the individual school to decide throughout the year if recess should occur outdoors or if the children can also stay indoors during recess due to weather/seasonal conditions. For middle and senior pupils, recess is typically characterised by free play both indoors and outdoors without any organised curriculum or teacher involvement except general schoolyard supervision.

### 2.2. Intervention

In October 2012, all public primary and lower secondary schools in Denmark (approximately 1800 schools) were invited to submit a proposal to re-design their schoolyards. From 106 submitted proposals, seven schoolyard projects were selected for refurbishment in February 2014. In 2015, the seven schoolyards were completed. Five of the seven schools agreed to participate in this follow-up study. They varied in geographic location (rural = 3, urban = 2), number of pupils enrolled (93–670 children), socioeconomic status (higher average income = 3, lower average income = 2), and size of the schoolyard per child (15–142 m^2^). 

The schoolyards were re-designed using a participatory bottom-up approach, engaging children in the design process. At some schools, architects were responsible for workshops with the children and at other schools, a process consultant was hired to facilitate the workshops. Gender, age, number and characteristics of children included in the process differed among the schools, as well as the time spent and techniques used with the children. At all the schools, the children were only part of the first step of the development process, providing ideas to the architects responsible for designing the schoolyards. Because of the participatory process, the outcomes were highly tailored and the design of the five re-designed schoolyards varied widely, however, common features included climbing walls, balance-bars, theatre/dancing-stages, skating areas, trampolines, hills, and ball game features. The total budget for each of the five schools ranged from 100,000 to 530,000 EUR [28]. 

### 2.3. Data Collection and Participants

As both daily PA [35,36] and recess PA [37] declines with age in childhood, we hypothesised that the perspectives of schoolyard designs would differ between girls of different ages, thus, we explored similarities and differences between the youngest (Grade 4; age 10–11) and oldest (Grade 6; age 12–13) girls in the target population of the re-designed schoolyards.

To recruit girls in our study, we made an announcement on the schools’ homepage addressed to parents having a daughter in Grade 4 and/or Grade 6) (potentially in total *n* = 235 girls). The announcement included information about the study and a phone number to which the parents had to send a text message if they allowed their daughter to participate. All girls from whom we had an electronic informed consent were invited to participate in an interview (*n* = 50 girls). Of these, 28 were in Grade 4 and 22 were in Grade 6, with a roughly equal split across schools (~10 per school).

Data were collected during a two-day visit to each school between April and June 2018 (Danish spring time). To obtain an in-depth insight into girls’ perspectives on their re-designed schoolyards, interviews were conducted using a photo-elicitation approach [38,39]. Prior to the interviews, the participating girls were instructed to take three photos during recess with their mobile phones; one photo describing where they typically spent their recess time, one photo of what they typically were doing in this space, and one photo showing who they typically spent time with. These photos were used during the interviews to stimulate dialogue on the girls’ reflections on their re-designed schoolyard and to trigger memories. This approach recognises children as the experts of their own experiences [40].

Forty-two interviews were conducted one-on-one with a researcher and four interviews were conducted in pairs since these girls felt more comfortable being interviewed with a classmate. All interviews were conducted during school hours in a separate room and lasted for approximately 30 min. Two experienced researchers conducted the interviews following a semi-structured interview guide developed to standardise interviews between subjects [41]. The structured questions helped to discuss the photos the girls took during recess and to collect important additional information not related to the photos. Questions in the interview guide included for example ‘Where are you during recess?’, ‘What are you doing there?’, ‘What else do you do during recess?’, ‘Who do you spend time with during recess?’ and ‘What do you think about your schoolyard?’ With the participants’ permission, the interviews were audio recorded using an iPad mini^®^.

### 2.4. Ethics

Prior to data collection, all the school principals and involved teachers provided approval to participate. Prior the interviews, all the girls were informed about the purpose and procedure of the study. They were also told that they could withdraw from the study at any time and that all the recordings were confidential. According to the Danish National Committee on Health Research Ethics, formal ethical approval was not required as the project was not a biomedical research project. The study and data management procedures were registered and approved by the Danish Data Protection Agency (2014-41-2801) and ISRCTN (ISRCTN17944407). 

### 2.5. Data Analysis

Data were analysed using pen profiles. This analysis procedure is a form of content analysis and is considered appropriate for representing analysis outcomes from large qualitative data sets via diagrams of composite key emergent themes [42,43]. 

After verbatim transcription of all interviews (in total 242 single-spaced pages of raw data in size 12, Times New Roman font), the transcriptions were read by the first author and quotations from the transcripts that referred to the physical environment influencing recess PA in the schoolyard were identified. All the quotations were coded manually and arranged under themes referring to physical environmental factors. The factors identified were reported as having either a positive (+ve) or a negative (-ve) influence on recess PA in the schoolyard and were coded accordingly. To enable comparisons between the two age groups, separate analyses were conducted for Grade 4 and Grade 6 girls. 

The analysis was initially presented to the research group by the first author by means of co-operative triangulation. The research group critically questioned the analysis being discussed until an acceptable consensus was reached. The first author then cross-examined the data in reverse to assure reliability of the data obtained [42]. For the purpose of these analyses, the number of times a specific factor was mentioned across all the interviews was presented in the diagrams. This provided an indication of the prevalence of each factor. If a participant mentioned the same factor at different times within the interview, the factor was only counted once per participant. A quotation was also extracted from the transcripts to further contextualise each factor. 

## 3. Results

Ten physical environmental factors influencing girls’ recess PA in the schoolyard were identified and grouped into three main themes: design, fixed facilities, and unfixed facilities. For the design theme, five factors were revealed: variety, accessibility, size, designated space and greenery. For fixed facilities, three factors were revealed: playground markings, active play facilities and sports facilities, and for unfixed facilities, two factors were revealed: play equipment and speakers. 

### 3.1. Design

Size refers to the perceived physical capacity of the schoolyard. It was reported to influence recess PA by ten Grade 4 and four Grade 6 girls. Nine Grade 4 girls and two Grade 6 girls reported that they had a spacious schoolyard and for that reason, they felt comfortable using it for PA. However, one Grade 4 girl and two Grade 6 girls felt crowded in the schoolyard during recess, which discouraged their use of the schoolyard (Figure 1). The three girls that perceived the schoolyard as crowded were all from the same school that had 15 m^2^ schoolyard per child. Children from the other four schools that had between 72–142 m^2^ schoolyard per child did not mention size. This suggests that the perceived schoolyard capacity seemed strongly related to the factual schoolyard size. 

In close relation to the perceived crowdedness in the schoolyard mentioned by three girls above, almost half (*n* = 23) of the interviewed girls reported that they preferred smaller designated spaces for PA secluded from the crowds and noise. Some of the girls preferred to use these spaces with a smaller group of girls whereas other girls preferred spaces solely allowed for their age group so that they were not disturbed by others. Almost half of the Grade 4 girls (*n* = 10) perceived that they had such spaces in the schoolyard, whereas three Grade 4 girls thought that these spaces were missing. Only two Grade 6 girls reported having access to a designated space in the schoolyard where they could stay without being disturbed by other children and eight Grade 6 girls wanted such a space. At some schools, they had designated spaces in the schoolyard, but the girls were not allowed to use the spaces because they were intended for another age group. The fact that girls expressed that they preferred smaller designated spaces seemed not only to be related to a feeling of crowdedness and having a small schoolyard, but also to the desire to have an area for themselves. Even at schools with plenty of space per child, girls were still attracted to smaller designated spaces.

The girls reported that greenery, such as grassy areas, trees, and bushes, was an important factor for encouraging recess PA and was often used as a designated space. Girls reported using greenery for activities such as hide and seek, den building, tree climbing, and role play. Some girls also expressed that they liked to play on grass because it did not hurt as much as asphalt if they fell over. Eleven of the Grade 4 girls were satisfied with the presence of greenery in the schoolyard, whereas five Grade 4 girls wanted their schoolyard to be greener. Fewer Grade 6 girls (*n* = 7) discussed the presence of greenery, with only three girls reporting that they were satisfied with the green features and four girls wanted more greenery in the schoolyard. Most of the girls who perceived a lack of greenery in the schoolyard were from the same school that had a green field area, but which lacked other natural features such as trees and bushes.

Eleven girls reported that accessibility influenced their recess PA in the schoolyard. Five girls (three Grade 4 and two Grade 6) explained that they used the schoolyard for PA because they perceived the schoolyard facilities to be located close to their classroom. In contrast, six Grade 6 girls perceived that the schoolyard facilities were placed at a distance from their classroom, hindering them in using the schoolyard for PA. All six Grade 6 girls that perceived the schoolyard facilities to be placed at a distance from their classroom attended the same school, which had a large schoolyard with the re-designed part of the schoolyard placed in a green area which was at a distance from the classrooms.

The variety of facilities and design in the schoolyard (e.g., surfaces and vegetation) supported opportunities for different activities and was an important factor for recess PA. Sixteen girls (ten Grade 4 and six Grade 6) thought that their schoolyard had a good variety of facilities and four girls (one Grade 4 and three Grade 6) were dissatisfied with the variety of the schoolyard features. At one school in particular, the girls were dissatisfied with the variety of facilities. At this school, the girls mentioned that most of the facilities in their schoolyard appealed to the younger children. 

### 3.2. Fixed Facilities

Four Grade 4 and five Grade 6 girls mentioned that playground markings, in particular foursquare markings, positively influenced their recess PA. Eight girls across the age groups were satisfied with the playground markings. However, one Grade 4 girl expressed a need for more foursquare markings since only four children could actively participate at one time and therefore, many children were waiting in line for their turn to play (Figure 2). At this school, a foursquare marking was removed as part of the re-design of the schoolyard.

Eight Grade 4 and Grade 6 girls reported that they liked to use sport facilities to play sports games during recess, such as soccer, basketball, and baseball. Only three Grade 4 girls and one Grade 6 girl were dissatisfied with the sport facilities available in the schoolyard. The girls who were satisfied with the sport facilities provided came from the same schools as the girls who were dissatisfied. The dissatisfied girls explained that they wanted more sport facilities than they already had (e.g., soccer and basketball facilities) so that they could play sport games without boys participating. 

Twenty Grade 4 girls reported that they preferred to be active in the schoolyard during recess by using active play facilities such as trampolines, obstacle courses, dancing scenes, and gymnastic equipment. The girls used these facilities to challenge themselves physically, such as doing tricks or performing. Ten of these girls were satisfied with the active play facilities provided in the schoolyard whereas the other 10 girls wanted more of these kinds of facilities. Active play facilities were mentioned less frequently by the Grade 6 girls (*n* = 8), with two girls being satisfied with the facilities provided and six girls wanting additional active play facilities in the schoolyard such as trampolines. The girls who were dissatisfied with the availability of active play facilities were from all five schools.

### 3.3. Unfixed Facilities

Seven girls (four Grade 4 and three Grade 6 girls) reported speakers for music to positively influence use of the schoolyard for dancing and creating a pleasant atmosphere. Also, music was mentioned to facilitated togetherness between groups of girls who normally did not play together. The girls who mentioned speakers were from the two same schools where they could connect their mobile phones to a speaker located in the schoolyard to play music (Figure 3).

Twenty-one girls (eleven Grade 4 and ten Grade 6 girls) reported that the availability of loose play equipment, such as different balls, baseball bats, jumping ropes and chalk, was important for encouraging recess PA in the schoolyard. However, most of these girls (seven Grade 4 and all ten Grade 6 girls) mentioned that there was not enough play equipment provided in the schoolyard. The girls who were dissatisfied with the availability of loose play equipment were from all five schools.

## 4. Discussion

This study aimed to investigate girls’ perceptions of physical environmental factors influencing their recess PA in re-designed schoolyards. Ten key factors within three main themes emerged: variety, accessibility, size, designated space, greenery, playground markings, active play facilities, sports facilities, play equipment, and speakers. The factors are further discussed below to obtain a greater understanding of what may be important to consider when (re)designing future schoolyards promoting girls’ recess PA.

### 4.1. Provision of Play Facilities Versus Sport Facilities

From the girls’ perspectives, it seemed important to distinguish between facilities for sport and facilities for play. Less than a third of the girls mentioned that they wanted facilities in the schoolyard for sport activities such as playing soccer, basketball, and baseball. In line with previous studies, the majority of the girls interviewed tended not to be interested in competitive sport-based activities because of low self-efficacy in sport and male dominance of sports games [10,12,13,14,15]. Despite this, the results showed that girls did want to be active in the schoolyard during recess but were more likely to mention activities that were not sport-related. For example, the girls reported that they liked challenging themselves and practicing gymnastic tricks and engaging in active role play, performance and games with a social focus. More than half of the interviewed girls reported that they preferred to play actively in the schoolyard by climbing, dancing and doing gymnastics, which require facilities that differ from traditional sporting facilities. The types of play facilities that the girls preferred were trampolines, obstacle courses, dancing scenes, gymnastic equipment and unfixed facilities that can be used for non-competitive active play, such as jumping ropes, speakers and chalk. Notably, variation in the facilities and equipment available to enable different kinds of activities seemed important to the girls. Previous studies have also shown that an increased number of fixed and un-fixed facilities in the schoolyard has a positive effect for PA among girls and boys of all ages [10,11]. In addition, greenery seemed important for girls as green areas appealed more to social play than competitive sport activities. In a recently published study by Raney et al. (2019), children, in particular girls, were less sedentary and participated in more collaborative play with peers in schoolyards where asphalt was replaced with grass, trees, and other natural features. In the study, the green zones were shown to provide new opportunities for engagement in non-sport activities like gymnastics, tag, and climbing activities [44], also mentioned to be preferred activities in the current study. 

The typical design of schoolyards reported in the literature to date has often had a strong sports focus through the provision of sports equipment and grounds. However, schoolyards do not usually provide the facilities necessary for dancing and gymnastics [10]. In our study, the re-designed schoolyards provided facilities for non-competitive active play. However, the girls reported mixed feelings concerning the adequacy of the play spaces and the equipment provided, which suggests that greater diversity and more active play facilities and equipment should be provided.

### 4.2. Creating a Motivational Atmosphere 

The social atmosphere in the schoolyard seemed very important to promote recess PA among girls. Almost half of the interviewed girls preferred to play in smaller designated spaces because of perceived crowds and noise in the schoolyard. This is consistent with other studies which showed that girls tended to play in smaller groups [45,46]. Moreover, music was mentioned as a way to create a positive atmosphere and encourage activity in the schoolyard. The girls from the two schools with a speaker in the schoolyard reported that the music facilitated togetherness between groups of girls who normally did not play together. Consistently with this finding, another schoolyard study found that girls liked performance facilitated by music and that group identities were enacted through performance with music [47]. 

Green areas were also often considered ‘cosy’ spaces by the girls. For example, a girl (quoted in Figure 2) mentioned that sounds from birds in their school garden area enhanced her use of the schoolyard. In a previous study, we found that children used these green areas to get away from others and for playing fantasy and self-invented games but that they were often lacking in schoolyards [48]. This finding resonates with Herrington and Brussoni (2015), who stated that the character and context of green environments can offer other types of stimuli than the traditional schoolyard environment [49]. Thus, when re-designing schoolyards, it seems important to consider how to generate atmospheres that girls feel comfortable in to encourage their recess PA. Providing music, green elements and smaller areas to reduce external influences (i.e., noise and other children) are examples of elements that can generate a comfortable atmosphere in schoolyards.

### 4.3. Designing for Varied Age Groups 

It was interesting that similar factors were reported to influence recess PA in the schoolyard by girls in the two different age groups, but some factors seemed more important to one age group than others. For example, active play facilities and greenery seemed most important to the Grade 4 girls. This age group appeared to enjoy playing in the schoolyard and explained that trees (for climbing) and grass (for soft fall) encouraged active play. The importance of play facilities and soft surfaces in the schoolyard was also found to be important for recess PA among 8–10 years-old girls [27]. However, as found in our study, previous studies have established that a change in play behaviour occurs among ages 10–12 and many traditional play facilities do not appeal to older children [10,50]. As such, these findings suggest that it can be more difficult to re-design activating schoolyards that appeal to children aged 13–15 years [51]. Consistently with this, our findings also revealed that some factors (e.g., designated space and accessibility) that were generally perceived as enablers by the Grade 4 girls were perceived as barriers by the Grade 6 girls. Accessibility, for example, was primarily reported to influence recess PA in the schoolyard by Grade 6 girls who perceived that the schoolyard facilities were at a distance from their classroom, negatively influencing their use of the schoolyard. This is consistent with another study that showed that Grade 2 and Grade 5 children had play settings in the schoolyard designed for their age by their classrooms, which the Grade 8 children did not have [52]. Christiansen et al. (2017) also found that location of facilities in close proximity to classrooms was important for use and the promotion of PA among 13–15 year-old children [51].

### 4.4. Strengths and Limitations 

The use of interviews based on photo-elicitation was valuable to capture the phenomenon of girls’ perceptions of the schoolyards during recess. Self-directed photos can capture ordinary interactions of children’s daily routines and uncover meaningful content areas that, from an adult viewpoint, might be overlooked [53]. It has been acknowledged that the inclusion of exploratory methodologies such as photo-elicitation and drawings may enhance the data gathered [27]. However, using a methodology based on the experiences of girls can be limiting since girls attending schools with few facilities may find it difficult to imagine or consider other possible designs and features. We experienced that the girls had clear opinions on how they wanted their schoolyard to be and echoing other child studies, they were delighted for their voices to be heard [30,54,55,56].

A limitation of using interviews is that the results are not generalisable due to the recruitment process and the fact that the data collection methods were not completely systematic [57]. However, our aim was to outline relevant key issues which might inform further research, and generalisability was not an expected attribute. 

Furthermore, the schoolyard interventions were developed using a participatory approach and were tailored to the needs of each school. Using a participatory design to develop tailored environmental interventions has proven to be an effective and viable approach [58]. This approach makes the results more difficult to interpret but having heterogeneity in the schoolyard design provides wider experiences with varying designs.

In the current study, we focused exclusively on physical environmental factors influencing girls’ recess PA in the schoolyard. We are aware that multiple other factors on the individual, cultural, and organisational level also can affect girls’ recess PA in the schoolyard [59,60,61]. However, through focusing on physical features that are important to girls’ recess PA in the schoolyard, this study will contribute to the design of future schoolyards.

## 5. Conclusions

This qualitative study offers important insights into girls’ perceptions of physical environmental factors that are important to be aware of when (re)designing schoolyards targeting girls’ recess PA. We found that the following factors were perceived to influence girls’ recess PA in the schoolyard: variety, accessibility, size, designated space, greenery, playground markings, active play facilities, sports facilities, play equipment, and speakers. Play facilities such as trampolines, obstacle courses, dancing scenes, and gymnastic equipment were favoured by girls over the more traditional sport facilities that are commonly provided. Furthermore, designated spaces, greenery and speakers were important features for girls to feel comfortable within the schoolyard. Similar factors were raised by girls in the two different age groups. However, our findings also revealed a transition from some factors generally perceived as enablers by the youngest girls (e.g., designated space and accessibility) to be perceived as barriers by the oldest girls. This highlights the complexity when trying to design schoolyards that cater to all age groups. 

### Practical Implications

We suggest a range of built environment actions to be considered when (re)designing schoolyards targeting girls’ recess PA. Our suggestions are:Provide a variety of facilities, surfaces, and vegetation in the schoolyardEmphasise fixed facilities such as trampolines, obstacle courses, dancing scenes, and gymnastic equipment to facilitate non-competitive play and social gamesProvide speakers for music and different unfixed play equipment such as balls, bats, skipping ropes, and chalkProvide a mixture of large and small secluded areas throughout the schoolyardConsider placing schoolyard facilities targeting the older girls near entry/exit points to school buildings

This study provided new insights into the perspectives of girls attending schools that underwent major schoolyard re-development. Further research in a larger sample investigating the complex interrelations between different age group of girls’ uses of different schoolyard environments is required.

## Figures and Tables

**Figure 1 ijerph-16-03508-f001:**
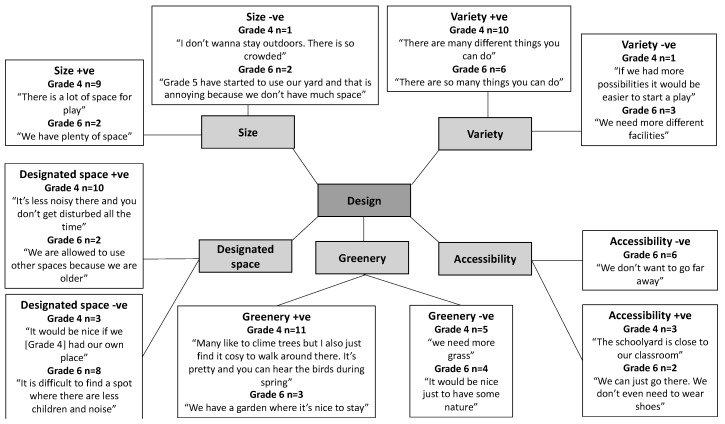
Factors related to design perceived to positively (+ve) and negatively (−ve) influence active schoolyard play by Grade 4 (*n* = 28) and Grade 6 girls (*n* = 22), respectively.

**Figure 2 ijerph-16-03508-f002:**
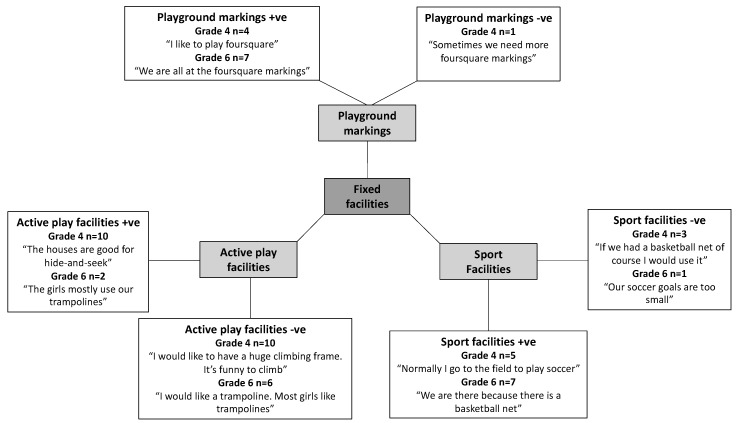
Factors related to fixed facilities perceived to positively (+ve) and negatively (-ve) influence active schoolyard play by Grade 4 (*n* = 28) and Grade 6 girls (*n* = 22), respectively.

**Figure 3 ijerph-16-03508-f003:**
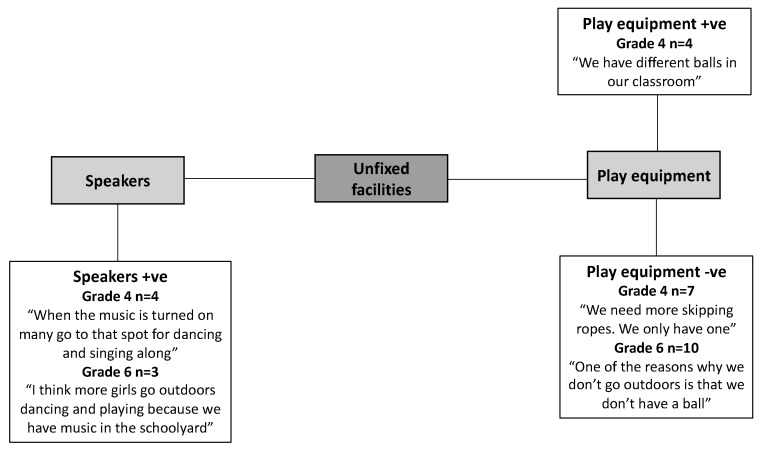
Factors related to unfixed facilities perceived to positively (+ve) and negatively (-ve) influence active schoolyard play by Grade 4 (*n* = 28) and Grade 6 girls (*n* = 22), respectively.

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
