# Peer review of "Designing Activating Schoolyards: Seen from the Girls’ Viewpoint"

_ijerph, 2019, doi:10.3390/ijerph16193508_

Round 1

Reviewer 1 Report

Thank you for the opportunity to review the current manuscript. This research follows a natural progression of the authors' work on the activating schoolyard study and provides a potentially meaningful contribution to the literature. The manuscript is also very well written. I have made more detailed comments below: 

Introduction 

The introduction is succinct, well-written, and appropriately cited. I do not believe any changes are necessary. 

Methodology

Overall the methodological approach for this study is appropriate. It would be helpful if the authors clarified the ontological/epistemological framing of the work. Additionally, more information about the participants would be helpful. Given the low consent rate, is there any information the authors have about the participants that might affect the results? For example, were the included students generally more active than the general population? Less active? Enjoy recess more or less? Such information would help contextualize the data in the results. 

Results 

Overall, the results would benefit from a more nuances analysis of the differences between various perspectives. Critically, in considering the application of findings for future interventions, understanding how satisfaction/dissatisfaction related to the actual physical environment, or differences in the physical environment will better help shape future practice. Often, the results are discussed as a function of how many participants noted one theme, without context of the actual physical environment, or if they environments in question are different (i.e., different schools, different play spaces). I have provided a few more specific comments below. 

Design. It seems there are some discrepancies in the text and Figure 1. For example Figure 1 suggests that 14 girls commented on the size of the playground (positive or negative) whereas the text notes that 23 girls commented on wanting smaller designated spaces. More clarification here is needed to explain the results. Moreover, Figure 1 has about 72 responses for the 50 participants. It would be important to know (a) if these responses represent all 50 participants or if there was a minority of participants that made up the responses in Figure 1 and (b) where there are differing view points, if these represent girls in different schools, and thus different experiences or if girls at the same school had differing opinions on the recess design. For example, some grade 4 girls noted the size of the playground as a positive, whereas others noted they did not have enough room. Is this a factor of different physical spaces or different cognitive appraisals of the space? Clarification here would be helpful. A similar point can be made in the text with different opinions on green space at recess. 

Fixed facilities/Unfixed Facilities. Similar to the comments above, I believe clarification on differing perspectives would be helpful here. 

Discussion

The authors note that, in line with previous research, girls are not interested in competitive sport based activities. While this is generally true, it is worth noting that some research seems to indicate that girls are interested in playing sport at recess, however they are excluded and eventually give up trying. A relevant citations might include (note: I believe the authors 2014 work also hints at some of this (citation #12 in the current paper, but I will leave that judgement to them): 

1.) Rodriguez-Navarro, Henar, Alfonso García-Monge, and Maria del Carmen Rubio-Campos. 2014. “The Process of Integration of Newcomers at School: Students and Gender Networking during School Recess.” International Journal of Qualitative Studies in Education27 (3): 349–363. doi:10.1080/09518398.2012.762472.

Author Response

Reviewer 1:

Thank you for the opportunity to review the current manuscript. This research follows a natural progression of the authors' work on the activating schoolyard study and provides a potentially meaningful contribution to the literature. The manuscript is also very well written. I have made more detailed comments below: 

--> Thank you for the comments. We believe that our revisions have further strengthened the paper. We have provided detailed responses to each of the comments below.

Introduction 

1. The introduction is succinct, well-written, and appropriately cited. I do not believe any changes are necessary. 

--> thank you for your positive comment

Methodology

2. Overall the methodological approach for this study is appropriate. It would be helpful if the authors clarified the ontological/epistemological framing of the work. Additionally, more information about the participants would be helpful. Given the low consent rate, is there any information the authors have about the participants that might affect the results? For example, were the included students generally more active than the general population? Less active? Enjoy recess more or less? Such information would help contextualize the data in the results. 

--> We have clarified the ontological/epistemological framing of the work, lines 67-76:

“In accordance with social constructionism [29], the intention of the study was to listen to the children and examine ways the schoolyard, as a social phenomenon, was perceived and institutionalisedby the girls. Allowing the girls to voice their perspectives is important in order to understand them and improve their PA [30]. A research perspective supportive of this approach is ‘the new social studies of childhood’ [31]. From the late 1990s onwards, this new approach sought to recognisechildren as active agents, competent social actors and individuals with rights [31]. However, in medical and health science research methods are predominantly adult-led and quantitative [32]. This may be valuable in identifying PA prevalence and associations. However, these methods do not provide contextual understanding and cannot readily explain why some children are more physically active than others [33, 34].”

We agree that the consent rate seems low, but it was not surprising. We “advertised” for girls (grade 4 and 6) to participate via the schools’ web site and asked the parents to return a consent form if they allowed their daughter to participate. We did not send a personal letter to the parents/girls and we did not send any reminders because fifty participants were a sufficient sample size for a qualitative study. We have clarified our recruitment procedure, lines 110-111:

“Prior to data collection, we invited to participation (girls in Grade 4 and Grade 6) via the schools’ web site (potentially in total n=235).”

Unfortunately, we are unable to determine whether specific girls participated and whether their activity levels were different as this was a different study (i.e. we could not link data between the original study and this).

Results 

3. Overall, the results would benefit from a more nuances analysis of the differences between various perspectives. Critically, in considering the application of findings for future interventions, understanding how satisfaction/dissatisfaction related to the actual physical environment, or differences in the physical environment will better help shape future practice. Often, the results are discussed as a function of how many participants noted one theme, without context of the actual physical environment, or if they environments in question are different (i.e., different schools, different play spaces). I have provided a few more specific comments below. 

--> Thank you for this interesting comment. We have now revised the result section and we have added more context and nuances about the actual physical environment. Please see lines 167-252.

4. Design. It seems there are some discrepancies in the text and Figure 1. For example, Figure 1 suggests that 14 girls commented on the size of the playground (positive or negative) whereas the text notes that 23 girls commented on wanting smaller designated spaces. More clarification here is needed to explain the results. Moreover, Figure 1 has about 72 responses for the 50 participants. It would be important to know (a) if these responses represent all 50 participants or if there was a minority of participants that made up the responses in Figure 1 and (b) where there are differing viewpoints, if these represent girls in different schools, and thus different experiences or if girls at the same school had differing opinions on the recess design. For example, some grade 4 girls noted the size of the playground as a positive, whereas others noted they did not have enough room. Is this a factor of different physical spaces or different cognitive appraisals of the space? Clarification here would be helpful. A similar point can be made in the text with different opinions on green space at recess. 

--> For the purpose of the analyses, the number of times a specific factor was mentioned across all interviews was presented in the diagrams. This provided an indicationof the prevalence of each factor. If a participant mentioned the same factor at different times within the interview, the factor was only counted once per participant, lines 156-158(i.e., some of the numbers in Figure 1 are low. This was based on the number of girls mentioning something, rather than the number of times it was mentioned). All 50 participants are represented in the responses and the schools are equally represented. We see size and designated space as two different factors. Size is about the perceived physical capacity of the schoolyard whereas designated space is about perceived place in space. We have tried to clarify your points throughout the design section, lines 167-215.

5. Fixed facilities/Unfixed Facilities. Similar to the comments above, I believe clarification on differing perspectives would be helpful here. 

--> We have also added more nuances of the differences between various perspectives in these two sections. Please see lines 216-252.

Discussion

6. The authors note that, in line with previous research, girls are not interested in competitive sport-based activities. While this is generally true, it is worth noting that some research seems to indicate that girls are interested in playing sport at recess, however they are excluded and eventually give up trying. A relevant citations might include (note: I believe the authors 2014 work also hints at some of this (citation #12 in the current paper, but I will leave that judgement to them): 

1.) Rodriguez-Navarro, Henar, Alfonso García-Monge, and Maria del Carmen Rubio-Campos. 2014. “The Process of Integration of Newcomers at School: Students and Gender Networking during School Recess.” International Journal of Qualitative Studies in Education27 (3): 349–363. doi:10.1080/09518398.2012.762472.

--> We have clarified this point, lines 261-267, and included the suggested reference

Reviewer 2 Report

Overall:

Manuscript is well-written and presents a pilot study that paves the way for additional studies related to schoolyard design and physical activity promotion. However, the implications of the results are currently overstated.

Introduction: See comments for conclusion related to literature review.

Methods:

Page 3, line 98, phrase should be “declines with”-This entire paragraph should be rewritten. Wording is repetitive

Authors should provide possible reasons for low recruitment

In order to fully understand why students viewed the schoolyard favorably or unfavorably, it is important to identify how many of the interview subjects also participated in the schoolyard design planning process and to what degree.

How did researchers account for the influence of the classmate in paired interviews?

How did researchers account for the influence of teachers during structured instructional time on how students spend their time during recess?

It is not clear how interviews were standardized between subjects. It is also not clear what is meant by interview guide or how reference #35 supports the interview method. Additional questions that were asked across participants would help clarify.

Results/Conclusions:

Because subject number was so low in each location and because location schoolyard square footage and design features were significantly different, it is difficult to make meaningful comparisons between grade levels. For example, when differences in the availability of space between ages is referenced in Figure 1, are these students from the same school sites? Suggest deleting all reference to age group comparisons throughout the manuscript unless more data can be presented that confirms it is an age-group difference and not a location, neighborhood, or income, or pre-design participation difference.

The study does a good job of highlighting factors that other researchers have previously identified as important for the promotion of physical activity participation by girls and the types of activities girls typically engage in during free play. However, it is not clear how the study advances our understanding of schoolyard design and physical activity more than any other previous study. A majority of these previous studies have significantly larger sample sizes and have used more objective measures of physical activity such as direct observation and GPS devices.

The gap in the literature is not which features promote physical activity by girls, but rather the extent to which those features need to be incorporated in future designs. The data presented does not adequately answer these questions.

Weaknesses in the study include subject number and a lack of a structured interview process. In addition to acknowledging these weaknesses and how they impact result interpretation, it is recommended that authors more explicitly acknowledge that the variety of schoolyard designs in the current study makes it difficult to combine results and extrapolate these results for any other schoolyard. As alluded to above, in addition to design features in the schoolyard itself, there is a mix of urban and rural neighborhoods, income levels, and schoolyard sizes represented in the study that can all have significant impacts on student perceptions and participation in physical activity. The potential impacts of these differences would be impossible to decipher given the sample size, not necessarily because of the study focus on physical features. It can be argued that this study alone may have little to no influence on the design of future schoolyards without additional larger and better controlled studies. It is recommended that authors provide specific recommendations about how the results from this study can be used to help design future studies.

Because boys were not also interviewed, it is difficult to determine whether their perceptions differed or if the suggestions provided by authors would truly meet the needs of all children.

In the conclusion section (p. 7, lines 233-236), it is not clear how the authors can confidently say that less than 1/3 of the girls wanted facilities for sport. Is it not possible that the girls interviewed simply did not think to mention the sport field if it wasn’t the location they photographed? Is it not also possible that some girls like sport fields and the trampoline equally? Greater details for the interview protocol may help the reader interpret results.

Conclusion section (p. 7, lines 248-253), authors are encouraged to complete another search of the recent literature and include reference to studies whose results have come to the same conclusions.

Conclusion section (p.8, lines 298-300), the current study does not provide any evidence that engaging students in all stages of the design process would change student perceptions three years post-design. It is suggested that authors delete this suggestion.

Author Response

Reviewer 2:

Overall: 

Manuscript is well-written and presents a pilot study that paves the way for additional studies related to schoolyard design and physical activity promotion. However, the implications of the results are currently overstated.

--> Thank you for the comments. We believe that our revisions have further strengthened the paper. We have provided detailed responses to each of the comments below.

Introduction: 

1. See comments for conclusion related to literature review.

--> See our response to this below (response to comment no. 14).

Methods: 

2. Page 3, line 98, phrase should be “declines with”-This entire paragraph should be rewritten. Wording is repetitive

--> We have rewritten the paragraph, lines 106-109.

3. Authors should provide possible reasons for low recruitment

--> We agree that the consent rate seems low, but it was not surprising. We “advertised” for girls (grade 4 and 6) to participate via the schools’ web site and asked the parents to return a consent form if they allowed their daughter to participate. We did not send a personal letter to the parents/girls and we did not send any reminder because we found fifty participants to be sufficient for a qualitative study. We have clarified our recruitment procedure, lines 110-111:

“Prior to data collection, we invited to participation (girls in Grade 4 and Grade 6) via the schools’ web site (potentially in total n=235).”

4. In order to fully understand why students viewed the schoolyard favorably or unfavorably, it is important to identify how many of the interview subjects also participated in the schoolyard design planning process and to what degree.

--> We agree that it would be interesting to look at how many and who of the interviewed subjects that took part in the schoolyard development process, but unfortunately, we are unable to assess this further as the research team were not involved in the development process. Therefore, we do not know whether they viewed the playground favorably or unfavorably due to whether the features they wanted were included (or not).

5. How did researchers account for the influence of the classmate in paired interviews?

--> From the transcriptions of the four interviews conducted with pairs it was clear that each participant answered differently to the questions asked and that they had individual opinions. So, we concluded that they did not influence each other in a way that was problematic to the analysis.

6. How did researchers account for the influence of teachers during structured instructional time on how students spend their time during recess?

--> Teachers did not take part in any aspect of the study and were not presented when the girls were informed about the study procedures.

7. It is not clear how interviews were standardized between subjects. It is also not clear what is meant by interview guide or how reference #35 supports the interview method. Additional questions that were asked across participants would help clarify.

--> We have added more questions from the interview guide and clarified how the interviews were standardized between subjects, lines 126-129:

”Two experienced researchers conducted the interviews followinga semi-structured interview guide developed to standardiseinterviews between subjects [41]. The structured questions helped to discuss the photos the girls took during recess, and to collect important additional information not related to the photos. Questions in the interview guide included: ‘Where are you during recess?’ ‘What are you doing there?’ ‘What else do you do during recess?’ ‘Who do you spend time with during recess?’ and ‘What do you think about your schoolyard?”

Results/Conclusions:

8. Because subject number was so low in each location and because location schoolyard square footage and design features were significantly different, it is difficult to make meaningful comparisons between grade levels. For example, when differences in the availability of space between ages is referenced in Figure 1, are these students from the same school sites? Suggest deleting all reference to age group comparisons throughout the manuscript unless more data can be presented that confirms it is an age-group difference and not a location, neighborhood, or income, or pre-design participation difference.

--> The subject numbers aren’t low given this is a qualitative study and in qualitative studies it is common to look at similarities and differences between groups like we have done and often with less participants than were included in our study. In the methods section we have highlighted that our study is a qualitative study and why this study is important, lines 73-76:

”in medical and health science research methods are predominantly adult-led and quantitative [32]. This may be valuable in identifying PA prevalence and associations. However, these methods do not provide contextual understanding and cannot readily explain why some children are more physically active than others [33, 34].”

Further, throughout the results section we have added more nuances of the differences between various perspectives. For example, lines 168-175:

“Size refers to the perceived physical capacity of the schoolyard. It was reported to influence recess PA by ten Grade 4 and four Grade 6 girls. Nine Grade 4 girls and two Grade 6 girls reported that they had a spacious schoolyard and for that reason they felt comfortable using it for PA. However, one Grade 4 girl and two Grade 6 girls felt crowded in the schoolyard during recess, which discouraged their use of the schoolyard (Figure 1). The three girls that perceived crowdedness in the schoolyard were all from the same school that had 15 m2 schoolyard per child. Children from the other four schools that had between 72-142 m2 schoolyard per child did not mention size? This suggests that perceived schoolyard capacity seemed strongly related to the factual schoolyard size.“

9. The study does a good job of highlighting factors that other researchers have previously identified as important for the promotion of physical activity participation by girls and the types of activities girls typically engage in during free play. However, it is not clear how the study advances our understanding of schoolyard design and physical activity more than any other previous study. A majority of these previous studies have significantly larger sample sizes and have used more objective measures of physical activity such as direct observation and GPS devices.

--> We believe that both quantitative and qualitative research is important to advance our understanding of schoolyard design and physical activity among girls. The quantitative studies may be valuable in identifying PA prevalence and identifying associations, but these methods do not provide contextual understanding and cannot readily explain the “why”. The current study gives an important insight into why girls choose the activities that they do and how they view the schoolyard from their perspective (i.e. we know what they do and where they go. This gives us the why in relation to physical activity – which is important information). In the re-submitted version we have elaborated the method sections to clarify the importance of this qualitative study. See our response to comment no. 8).

10. The gap in the literature is not which features promote physical activity by girls, but rather the extent to which those features need to be incorporated in future designs. The data presented does not adequately answer these questions.

--> We don’t agree here – there isn’t a big literature on what features promote activity in girls, but we agree that we need to know more about how these designs are incorporated into future designs. We have included this point in the conclusion, lines 376-379:

“The study has provided new insights into the perspectives of girls attending schools that underwent major schoolyard re-development. Further research in a larger sample investigating the complex interrelations between different age group of girls’ uses of different schoolyard environments is required.”

11. Weaknesses in the study include subject number and a lack of a structured interview process. In addition to acknowledging these weaknesses and how they impact result interpretation, it is recommended that authors more explicitly acknowledge that the variety of schoolyard designs in the current study makes it difficult to combine results and extrapolate these results for any other schoolyard.As alluded to above, in addition to design features in the schoolyard itself, there is a mix of urban and rural neighborhoods, income levels, and schoolyard sizes represented in the study that can all have significant impacts on student perceptions and participation in physical activity. The potential impacts of these differences would be impossible to decipher given the sample size, not necessarily because of the study focus on physical features. It can be argued that this study alone may have little to no influence on the design of future schoolyards without additional larger and better controlled studies. It is recommended that authors provide specific recommendations about how the results from this study can be used to help design future studies.

--> We do not agree that subject number and lack of structured interview process is a weakness in the study. We used a semi-structured interview process clarified, lines 126-139 and interviewing 50 girls is a high number of subjects in a qualitative study.

We do agree that the variety of schoolyard designs in the current study makes it difficult to interpret the results but having heterogeneity in the schoolyard design actually strengthens the study as it provides wider experiences with varying playground designs. We have discussed that in the strengths and limitations section, lines 341-345:

“Further, the schoolyard interventions were developed using a participatory approach and was tailored to the needs of each school. Using a participatory design to develop tailored environmental interventions has proven to be an effective and viable approach [57]. This approachmakes the results more difficult to interpret but having heterogeneity in the schoolyard design provides wider experiences with varying designs.”

Further, to (re)design schoolyards targeting girls’ recess PA we have suggested our findings to be elaborated in a larger study. See abstract, line 23 and conclusion, lines 378-380:

“Further research in a larger sample investigating the complex interrelations between different age group of girls’ uses of different schoolyard environments is required.”

12. Because boys were not also interviewed, it is difficult to determine whether their perceptions differed or if the suggestions provided by authors would truly meet the needs of all children.

--> True, we do not know if our suggestions would meet the needs of all children and neither it was the aim of the study. If we truly want to address the gap in boys and girls PA levels, we believe that a special attention on the girls is needed.

13. In the conclusion section (p. 7, lines 233-236), it is not clear how the authors can confidently say that less than 1/3 of the girls wanted facilities for sport. Is it not possible that the girls interviewed simply did not think to mention the sport field if it wasn’t the location they photographed? Is it not also possible that some girls like sport fields and the trampoline equally? Greater details for the interview protocol may help the reader interpret results.

--> True, we have corrected the sentence to “mentionedthat they wanted…”, line 262. And you are right that some girls mentioned both sport and play facilities, but overall girls mentioned play facilities more than sport facilities. We have elaborated on our description of the interview protocol procedure, lines 126-132.

14. Conclusion section (p. 7, lines 248-253), authors are encouraged to complete another search of the recent literature and include reference to studies whose results have come to the same conclusions.

--> We have completed another search of recently published literature on the topic of our paper but unfortunately, we did not find any suitable studies. If the reviewer knows any studies suitable to strengthen our conclusions, we would be very happy to include them.

15. Conclusion section (p.8, lines 298-300), the current study does not provide any evidence that engaging students in all stages of the design process would change student perceptions three years post-design. It is suggested that authors delete this suggestion.

--> True, we have deleted this suggestion.

Round 2

Reviewer 2 Report

The revised manuscript is much improved. Specifically, the expanded explanation of study purpose and design theory on page 2 and the more detailed descriptions provided throughout the results section have elevated manuscript quality and reader relatedness. A couple of additional minor edit suggestions are presented below:

1) Page 3, lines 110-111-wording is a bit awkward. In addition, it is still not clear what is meant by “invited to participation via the school’s website”. Was there a link on the school’s homepage that parents would have to click on to access and then print an informed consent or was there an announcement directly on the homepage about the study? What details were included in the advertisement? Is there any concern that the students who returned consents come from families who are actively engaged in school functions and families who do not regularly access the school’s website (e.g. lower income families which are less likely to accumulate the recommended PA levels) would never know about the study? An accurate interpretation of results would require additional details about study participants. Is there any explanation for the dramatic difference in the percentage of students participating from each school site (10 out of 93 vs 10 out of 670)? It is recommended that authors make specific note in the Strengths and Limitations section about the inability of researchers to verify that interviews were conducted from a representative sample. 

2) Page 9, lines 322-330-I reiterate that because authors cannot identify which study participants provided input to the architects, it is not clear if listening to the children throughout the design process would be more or less beneficial to increasing PA in girls. This is a new idea presented in the discussion that is not related to the data collected and therefore should not be included in the manuscript.

3) I agree with the authors that it is valuable to combine information from quantitative studies with qualitative studies that assess student perceptions. Student perceptions have the ability to confirm conclusions/hypotheses made by others. Indeed, many of the themes identified by authors and represented in the figures are contextual play factors that have been presented previously. For example, other studies have concluded that girls prefer specific activity modes in specific areas with specific design features in the schoolyard and that play preferences change with age. Recognition that the contextual play factors influencing PA by girls of different ages have been previously identified through observational studies is important. Here is just a small sample of these previous studies:

Pagels, P., Raustorp, A., De Leon, A. P., Mårtensson, F., Kylin, M., & Boldemann, C. (2014). A repeated measurement study investigating the impact of school outdoor environment upon physical activity across ages and seasons in Swedish second, fifth and eighth graders. BMC Public Health14, 803. doi:10.1186/1471-2458-14-803

Czalcynska-Podolska, M. (2014). The impact of playground spatial features on children’s play and activity forms: An evaluation of contemporary playgrounds’ play and social value. Journal of Environmental Psychology, 38, 132-142. doi: 10.1016/j.jenvp.2014.01.006

Raney, M. A., Hendry, C. F., Yee, S. A. (2019). Physical activity and social behaviors of urban children in green playgrounds. American Journal of Preventive Medicine, 56, 522-529. doi: 10.1016/j.amepre.2018.11.004

Author Response

Response to reviewer 2:

The revised manuscript is much improved. Specifically, the expanded explanation of study purpose and design theory on page 2 and the more detailed descriptions provided throughout the results section have elevated manuscript quality and reader relatedness. A couple of additional minor edit suggestions are presented below:

--> Thank you for the positive statements and your comments. We have now made revisions and believe that the revisions have further strengthened the paper. We have provided detailed responses to each of the comments below.

1) Page 3, lines 110-111-wording is a bit awkward. In addition, it is still not clear what is meant by “invited to participation via the school’s website”. Was there a link on the school’s homepage that parents would have to click on to access and then print an informed consent or was there an announcement directly on the homepage about the study? What details were included in the advertisement? Is there any concern that the students who returned consents come from families who are actively engaged in school functions and families who do not regularly access the school’s website (e.g. lower income families which are less likely to accumulate the recommended PA levels) would never know about the study? An accurate interpretation of results would require additional details about study participants. Is there any explanation for the dramatic difference in the percentage of students participating from each school site (10 out of 93 vs 10 out of 670)? It is recommended that authors make specific note in the Strengths and Limitations section about the inability of researchers to verify that interviews were conducted from a representative sample. 

--> We have included more information about our advertisement procedure in lines 110-114:

“To recruit girls in our study, we made an announcement on the schools’ homepage addressed to parents having a daughter in Grade 4 and/or Grade 6) (potentially in total n=235 girls). The announcement included information about the study and a phone number to which the parents had to send a text message if they allowed their daughter to participate. All girls, from whom we had an electronic informed consent were invited to participate in an interview (n=50 girls).”

In the strengths and limitations section we have included the following section, lines 341-344:

“A limitation of using interviews is to secure a representative sample and that the results are not generalisabledue to the recruitment process and the fact that the data collection methods were not completely systematic (Crabtree & Miller, 1999). However, our aim was to outline relevant key issues which might inform further research, and generalisabilitywas not an expected attribute.”

2) Page 9, lines 322-330-I reiterate that because authors cannot identify which study participants provided input to the architects, it is not clear if listening to the children throughout the design process would be more or less beneficial to increasing PA in girls. This is a new idea presented in the discussion that is not related to the data collected and therefore should not be included in the manuscript.

--> We agree and have deleted this section.

3) I agree with the authors that it is valuable to combine information from quantitative studies with qualitative studies that assess student perceptions. Student perceptions have the ability to confirm conclusions/hypotheses made by others. Indeed, many of the themes identified by authors and represented in the figures are contextual play factors that have been presented previously. For example, other studies have concluded that girls prefer specific activity modes in specific areas with specific design features in the schoolyard and that play preferences change with age. Recognition that the contextual play factors influencing PA by girls of different ages have been previously identified through observational studies is important. Here is just a small sample of these previous studies:

Pagels, P., Raustorp, A., De Leon, A. P., Mårtensson, F., Kylin, M., & Boldemann, C. (2014). A repeated measurement study investigating the impact of school outdoor environment upon physical activity across ages and seasons in Swedish second, fifth and eighth graders. BMC Public Health14, 803. doi:10.1186/1471-2458-14-803

Czalcynska-Podolska, M. (2014). The impact of playground spatial features on children’s play and activity forms: An evaluation of contemporary playgrounds’ play and social value. Journal of Environmental Psychology, 38, 132-142. doi: 10.1016/j.jenvp.2014.01.006

Raney, M. A., Hendry, C. F., Yee, S. A. (2019). Physical activity and social behaviors of urban children in green playgrounds. American Journal of Preventive Medicine, 56, 522-529. doi: 10.1016/j.amepre.2018.11.004

--> Thank you for sharing the above publications. We have read them with interest and added two of them in the discussion section of our paper, lines 279-284 and 326-328. We did not include the paper by Czalcynska-Podolska even they had a similar focus since they did not do separate analysis on boys and girls
